# Twelve-Week Safety and Potential Lipid Control Efficacy of Coffee Cherry Pulp Juice Concentrate in Healthy Volunteers

**DOI:** 10.3390/nu15071602

**Published:** 2023-03-25

**Authors:** Numphung Rungraung, Niramol Muangpracha, Dunyaporn Trachootham

**Affiliations:** Institute of Nutrition, Mahidol University, Nakorn Pathom 73170, Thailand; numphung.run@mahidol.ac.th (N.R.); niramol.mua@mahidol.ac.th (N.M.)

**Keywords:** coffee cherry pulp juice concentrate, polyphenols, randomized control trial, safety, healthy volunteers, novel food, blood lipid, blood sugar

## Abstract

Coffee cherry pulp, a major waste product from coffee manufacturing, contains polyphenols with antioxidant activity. However, its clinical safety and health benefits are unclear. This randomized, double-blinded, placebo-controlled trial evaluated the safety and potential efficacy of coffee cherry pulp juice concentrate. A total of 61 participants were randomly divided into a study group (*n* = 30), receiving the juice, and a control group (*n* = 31), receiving a placebo drink of 14 g twice daily for 12 weeks. Adverse symptoms, changes in body weight, hematological and biochemical parameters, vital signs, and heart function were evaluated using subject diaries, interviews, blood and urine tests, and electrocardiograms. The results showed no intervention-related adverse events. Body weight, liver, renal function, complete blood counts, blood glucose, urinalysis, and electrocardiograms were not significantly altered throughout the study. Consuming the juice for at least 8 weeks significantly decreased cholesterol and LDL levels. The glucose levels were maintained significantly better than those of the placebo group. The findings suggest that continuously consuming 28 g/day of coffee pulp juice concentrate for 12 weeks is safe in healthy volunteers. Future studies could employ a dose of ≤28 g/day to investigate the efficacy of this novel food, especially for preventing dyslipidemia and diabetes.

## 1. Introduction

Coffee pulp, which refers to the skin and the pulp of coffee fruit, is the main residue produced during coffee production [1,2]. This by-product contains bioactive compounds, especially polyphenols, such as chlorogenic acid (CGA), caffeine, catechin, epicatechin, chlorogenic acid, and rutin [3,4]. These compounds exhibit functional effects, such as lowered cholesterol, improved liver steatosis, and reduced hyperlipidemia, as well as antioxidant and anti-inflammation effects, and improved cognitive function [5,6,7,8,9]. Coffee cherry skin and pulp products are considered a novel food, according to the European Union (EU)’s and Thailand’s Novel Food legal regulations [10,11]. Novel food requires a scientific assessment of safety before submitting its label to regulatory organizations, such as the European Food Safety Authority (EFSA) or the Thai Food and Drug Administration (Thai FDA), for approval before use. Furthermore, information on safe doses in humans is required in designing clinical trials for the efficacy of functional novel food. Therefore, toxicological studies in animals and clinical safety studies in healthy people are crucial. The safety of whole coffee fruits has been reported in several studies. Heimbach et al. reported a subchronic toxicity study of Coffeeberry^®^, the ethanolic extract of the whole coffee fruit. The result showed that 3446 and 4087 mg/kg BW/day for male and female rats, respectively, did not cause any adverse effects [12]. Up to 300 mg of Coffeeberry^®^ per serving in conventional foods has been listed as Generally Recognized as Safe (GRAS) Notice (GRN) No. 868 [13]. One clinical study also showed that supplementation with whole coffee fruit powder at 800 mg per day for 28 days in college athletes increased antioxidant capacity with no adverse event effects [14]. Nevertheless, the safety profile of products from coffee pulp, which is the part that contains the richest levels of phytochemicals, is unknown. Since components of coffee pulp, such as caffeine and tannins, or contaminants, such as mycotoxins, can pose a safety concern [15], it is important to evaluate the long-term safety of coffee pulp products.

Coffee pulp can be processed into various food commodities, such as jam, juice, concentrate, jelly, flavoring, and alcoholic beverages [2,16]. Pressing of coffee pulp yields 14° Brix juice [17], which can evaporate further to make a coffee pulp juice concentrate. Recently, we reported a dose-escalation study of coffee cherry pulp juice concentrate. Up to 14 g twice daily (28 g per day) is the maximum tolerated dose for two-week consumption of coffee cherry pulp juice concentrate in healthy participants [18]. However, the safety and health benefits of its long-term consumption remained unknown, and the placebo effect has not been studied. Thus, this randomized, double-blind, placebo-controlled study aimed to evaluate the safety and potential efficacy of coffee cherry pulp juice concentrate continuous intake for 12 weeks.

## 2. Materials and Methods

### 2.1. Ethical Aspects and Setting

The protocol of this study (MU-CIRB 2021/275.1905) was approved by Mahidol University Central. Institutional Review Board (MU-CIRB), approval number COA No.MU-CIRB2020/150.2406. This research was performed according to the International Council for Harmonization of Technical Requirements for Pharmaceuticals for Human Use Good Clinical Practice (ICH-GCP) and the Declaration of Helsinki. Informed written consent was obtained from each participant before the study commenced. The protocol was registered in the Thai Clinical Trial Registry (TCTR20230103004) and can be accessed at https://www.thaiclinicaltrials.org/show/TCTR20230103004 (accessed on 19 March 2023).

### 2.2. Study Design, Blinding, Random Allocation, and Concealment

A randomized, double-blind, placebo-controlled trial was used. Participants who passed the screening were randomly allocated into two groups, including the study (receiving coffee cherry pulp juice concentrate) and the control (receiving a placebo drink) groups. All participants received unlabeled sachets of the assigned product so that they were blinded from their group identity. Furthermore, the laboratory scientist and statistical analyzer were also blinded from the assignment list.

### 2.3. Participants

The inclusion criteria were as follows: aged 20–55 years old, healthy with no systemic diseases, body mass index (BMI) ≤ 30 kg/m^2^, acceptable blood chemistry, i.e., hemoglobin 11–16 gm/dL for women and 12–18 g/dL for men, WBC 4000–10,000 cells/mm^3^, fasting plasma glucose ≤ 125 mg/dL, ALT ≤ 60 U/L, eGFR ≥ 90 mL/min/1.73 m^2^, acceptable blood pressure and heart rate, i.e., systolic blood pressure between 90 and 140 mmHg, diastolic blood pressure between 60 and 90 mmHg, and heart rate between 60 and 100 beats/min. Blood cholesterol > 200 mg/dL is common even in healthy populations. To reflect the status of the general population and ensure the feasibility of this study, we included participants with total cholesterol ≤ 280 mg/dL and with total cholesterol:HDL ratio ≤ 5:1. The cut-off values were set based on previous studies for a significantly lower risk of mortality in the general population [19,20].

Exclusion criteria were as follows: COVID-19-infected, allergic to coffee, caffeine, or herbal extracts, unable to come for follow-up every 4 weeks, planning to become pregnant within 3 months, pregnant or breastfeeding, regularly consuming drugs/herbs/dietary supplements, drinking alcohol > 14 drinks/week for men or 7 drinks/week for women, smoking > 10 cigarettes/day, blood clotting problems such as idiopathic thrombocytopenia, or having conditions not suitable for consuming caffeine-containing products such as insomnia, uncontrolled hypertension, glaucoma, severe gastritis, dyslipidemia, osteoporosis, irritable bowel syndrome, vitamin B1 deficiency, or menopausal syndrome. All participants provided their written informed consent before data collection.

### 2.4. Sample Size and Power

The sample size of the present study was calculated using G-power V.3.1.9.4 (Heinrich-Heine-Universität Düsseldorf, Düsseldorf, Germany) [21]. A theoretical large effect size value of 0.8 for the comparison of two independent means (the unpaired *t*-test) was used. With a power of 0.8 and a significance level of 0.05, the calculation of the sample size yielded the result of at least 26 people per group (Appendix A). Assuming a dropout rate of 15%, the sample size was set at 30 people (*n* = 30) in each group, and the total sample size was 60.

### 2.5. Intervention and Materials

Our previous study showed that consuming 14 g of coffee cherry pulp juice concentrate twice daily (28 g per day) was safe [18]. Thus, in this study, participants were asked to consume a sachet of 14 g coffee cherry pulp juice concentrate or a placebo twice a day (one before breakfast and another before bed) for 12 weeks. Each 14 g sachet contained polyphenols 62.5–69.5 milligram gallic acid equivalents (mg eq GA) with a total antioxidant of 1392.5–1539.5 micromoles of Trolox equivalents (μmoles TE). Thus, participants in the study group received polyphenols 126–139 mg eq GA with a total antioxidant content of 2785–3079 μmoles TE per day.

Both the coffee cherry pulp juice concentrate and the placebo were manufactured in a good manufacturing practice (GMP)-certified plant by MiVana Co., Ltd., Samut Prakan, Thailand. Coffee cherry pulp juice concentrate was made from the frozen coffee pulp (skin and pulp) of *Coffee arabica* L. variety Catimor, which is the by-product of coffee bean processing. Coffee pulp juice with 9–13° Brix was made by squeezing the thawed coffee pulp. Then, the juice was vaporized at 55 °C for 16 h, yielding the concentrate at 50 °Brix. The juice concentrate was packed in a retort pouch (14 g sachet) and sterilized by water spray retort processing at 100 °C for 12 min. The product passed the microbiological and contaminant safety standards following the Notification of the Ministry of Public Health (No.355) B.E. 2556 (2013) for the control of food in hermetically sealed containers [22]. Aflatoxin, arsenic, cadmium, lead, mercury, and tin were not detected. It should be noted that all products used in this study were produced in the same manufacturing batch and the manufacturer also performed quality control checks to ensure minimal variation between different batches. Per 100 g of the product, there is 0.29–0.33% caffeine, 2.7–3.00 g dietary fiber, 3.3–3.7 g protein, 0.18–0.20 mg vitamin E, 34.2–37.8 g sugar, 0.17–0.19 mg zinc, and 0.716–0.796 mg copper. Total polyphenol content is used for the quality control of active compounds, with the range of 449.9–497.9 mg eq GA per 100 g of product. The water activity of the product at 25 °C is 0.846–0.926. The total antioxidant activity measured by ORAC assay is 9947.13–10,995.13 μmoles TE per 100 g product. The placebo was made using water, fructose syrup, sodium citrate, malic acid, brown color powder, and carboxymethylcellulose to match the coffee cherry pulp juice concentrate for weight, taste, color, flavor, and acidity (equal pH of 4.5). The product was filled in and sterilized by water spray retort processing at 100 °C for 12 min, similar to those of the coffee cherry pulp juice concentrate. The placebo products passed the microbiological and contaminant safety standards, similar to those of coffee cherry pulp juice concentrate. To ensure the similarity between the placebo and the juice, a group of trained panelists tasted both products and confirmed that the taste of a placebo product was similar to that of the juice concentrate, which is both sweet and sour (similar to prune juice). Additionally, the brix and pH were measured to ensure similar properties of both products.

### 2.6. Study Procedure

This work was performed at the Institute of Nutrition, Mahidol University. Participants who passed the screening were randomly assigned into one of two groups. Each participant consumed the assigned product, a sachet of 14 g/day, twice daily before breakfast and before bed (for a total of 28 g/day) for 12 weeks. The sachets of both the coffee cherry pulp juice concentrate and the placebo are ready-to-eat. All participants were asked to consume the products directly from the sachets by using the provided disposable straws. They were instructed to record their daily consumption of the assigned products in their subject diaries. Additionally, they kept weekly diet records for three days per week (two weekdays and one weekend day) to describe other food they consumed throughout the study. The investigator checked the records in every follow-up visit to ensure the consistency of their consumption pattern. All participants were asked to maintain their usual physical activities throughout the study.

During the consumption period, participants were instructed to follow up every 4 weeks. Follow-up was also conducted 2 weeks after stopping the intervention to assess safety. Signs of adverse events were measured using a blood test, urinalysis, body weight, vital signs, and electrocardiogram (EKG) at baseline (week 0) and at weeks 4, 8, 12, and 14 (2 weeks after stopping the intervention). Symptoms of adverse events were monitored daily by self-records in subject diaries and interviews by a researcher at the follow-up visits. The conceptual framework is shown in Figure 1.

### 2.7. Outcome Measurement

The safety and potential efficacy of coffee cherry pulp juice concentrate compared to a placebo was assessed using body weight, vital signs (blood pressure and pulse rate), and clinical laboratory tests including fasting plasma glucose (FPG), liver function (aspartate transaminase (AST), alanine aminotransferase (ALT), and total bilirubin), renal function (blood urea nitrogen (BUN) and creatinine (CR), lipid profile (total cholesterol, high-density lipoprotein (HDL), low-density lipoprotein (LDL), and total triglycerides), complete blood count (CBC), urinalysis tests, and electrocardiograms (EKGs). Throughout the study, adverse symptoms were recorded by the participants in subject diaries, and participants were interviewed by a researcher at every follow-up visit. Body weight was determined using a bioelectrical impedance machine, TANITA BC-730 (Tanita Corporation, Tokyo, Japan). Blood pressure and pulse rate were measured using a Digital Blood Pressure Monitor OMRON HEM-7156A (OMRON HEALTHCARE Co., Ltd., Kyoto, Japan). Electrocardiograms (EKG) were measured using a portable ECG monitor Contec PM10 (Contec Medical Systems Co., Ltd., Qinhuangdao, Hebei, China). FBS, AST, ALT, BUN, CR, and lipid profile were measured using a semi-automatic biochemistry analyzer Layto RT9200 (Rayto Life and Analytical Sciences Co., Ltd., Shenzhen, China). CBC was measured using a Sysmex XN-10™ Automated Hematology Analyzer (Sysmex Corporation, Kobe, Japan).

### 2.8. Statistical Analysis

Numerical and categorical data were summarized as mean value ± standard deviation and frequencies (in %), respectively. Baseline characteristics between study and control groups were compared using Fisher’s exact tests for categorical data and unpaired *t*-tests for numerical data. Changes in each parameter over time within the same group were analyzed by using the Kruskal–Wallis test. The differences in parameter changes over time between the study and control groups were analyzed using mixed-effect analysis followed by Sidak’s multiple comparison test. A *p*-value of less than 0.05 was considered statistically significant.

## 3. Results

### 3.1. Participant Flow Chart

This study was conducted from September 2021 to May 2022. Figure 2 shows the Consolidated Standards 203 of Reporting Trials (CONSORT) participant flow diagram. There were 71 participants initially screened for eligibility, and 61 of them passed the screening. At first, 60 participants (*n* = 30 for each group) were randomly assigned to the study and placebo groups. Then, there were two participants in the study group and four participants in the control group who dropped out before completing the study. Since the control group had more dropouts than the study group, to account for that imbalance, we recruited one more participant into the control group. The data of all participants were included in the intention-to-treat analysis. The study group contained complete data of all time points from 28 participants and data with some missing values from 2 participants. The control group contained complete data of all time points from 27 participants and data with some missing values from 4 participants. Mixed effect analyses were used for statistical tests.

### 3.2. Baseline Demographic, Clinical Chemistry, and Hematology Characteristics of the Participants

The baseline characteristics of participants are summarized in Table 1 and Table 2. No significant inter-group differences were found for gender, age, BMI, congenital disease presence, smoking status, alcohol intake, systolic blood pressure, diastolic blood pressure, or pulse rate (*p* > 0.05) (Table 1). No hematological parameters were significantly different between the study and placebo groups (Table 2). However, several blood chemistry values including fasting plasma glucose, total cholesterol, HDL, LDL, creatinine, eGFR, AST, and ALT in the study group were slightly but statistically significantly higher than in the placebo (Table 2). Therefore, when comparing the difference between groups, the % baseline of each parameter was used instead.

### 3.3. Adverse Symptoms

No adverse symptoms were observed in either the study group or the control group during the study (Table 3).

### 3.4. Changes in Body Weight

The average body weight was not significantly changed throughout the study in the study group (Figure 3A), with no significant differences between groups (Figure 3B).

### 3.5. Changes in Fasting Plasma Glucose

After 4 weeks of consumption of coffee cherry pulp juice concentrate, the average fasting plasma glucose was significantly decreased compared to the baseline (*p* < 0.01). However, it returned close to the baseline at weeks 8 and 12, and it was significantly decreased compared to the baseline after stopping coffee cherry pulp juice concentrate intake for 2 weeks (*p* < 0.05) (Figure 4A). There were significant differences between the two groups (*p* < 0.0001), and the result shows that the study group maintained fasting plasma glucose significantly better than the placebo group (Figure 4B).

### 3.6. Changes in Blood Lipid Profile

The average total cholesterol level was significantly decreased after 8- and 12-week consumption of coffee cherry pulp juice concentrate (*p* < 0.001 and *p* < 0.05, respectively). Additionally, even after stopping the intervention for 2 weeks, the average total cholesterol was still significantly lower than the baseline (*p* < 0.05) (Figure 5A). Compared to the placebo group, the study group had significantly reduced levels of total cholesterol starting from week 4 (*p* < 0.0001) (Figure 5B). Similarly, the average level of LDL was significantly decreased after 8- and 12-week consumption of coffee cherry pulp juice concentrate (*p* < 0.01 and *p* < 0.05, respectively). Additionally, the average level of LDL was still significantly lower than the baseline even after stopping the intervention for 2 weeks (*p* < 0.01) (Figure 5C). No significant change was observed in the placebo group. Compared to the placebo group, the study group had significantly reduced levels of LDL starting from week 4 (*p* < 0.0001) (Figure 5D). During the 12-week consumption period, the levels of HDL and triglyceride did not significantly change, but an increasing trend was observed two weeks after stopping the intervention (Figure 6A,C). Compared to the placebo, HDL and triglyceride in the study group were increased significantly after stopping the intervention for 2 weeks (*p* < 0.0001 and *p* < 0.05, respectively) (Figure 6B,D). The average increased triglyceride level was still within the normal range.

### 3.7. Changes in Kidney Function

As shown in Figure 7A,B, the average BUN level in the study group was not significantly changed throughout the study and there were no significant differences from the placebo group. In contrast, creatinine was significantly decreased after 12-week consumption of coffee cherry pulp juice concentrate (*p* < 0.001), and after stopping the intervention for 2 weeks (*p* < 0.001), compared to the baseline (Figure 7C). Starting from 4 weeks of intervention until the end of the study, there was a significant difference in creatinine levels between the study and placebo groups (*p* < 0.0001). Whereas the placebo group had a significant increase in creatinine (but still within the normal range), the study group had a reduced creatine level (Figure 7D).

### 3.8. Changes in Liver Enzymes

The average levels of ALT and AST in the study group were not significantly altered (Figure 8A,C). While there was no difference in ALT between the study and placebo groups, the average AST level of the placebo group was significantly higher than that of the study group at 2 weeks after stopping the intervention (*p* < 0.001) (Figure 8B,D). The average increased AST level was still within the normal range.

### 3.9. Changes in Hematological Parameter

No hematological parameters, such as hemoglobin, hematocrit, and WBC, were significantly changed in either group, and there was no significant difference between groups (Figure 9A–F).

### 3.10. Changes in Urinalysis

Most participants in both groups maintained normal properties of urine throughout the study. As shown in Table 4, there were some participants with abnormalities at certain time points. Participants with slightly cloudy urine, RBC, WBC, and squamous epithelial cells were experiencing menstruation at the studied time point. Participants with protein in their urine had normal renal function values but had drunk less water at the studied time point.

### 3.11. Changes in EKG

The EKG profile was not significantly altered throughout the study after consuming the coffee cherry pulp juice concentrate (Figure 10A). The percentages of participants with normal electrocardiograms in both groups were not different throughout the study (Figure 10B).

## 4. Discussion

Transforming waste from the food industry into functional food has become an emerging area of research and innovation [23]. Not only does it help the environment, but it also creates new plant-based food without the need to grow a new plant, which facilitates food security [23]. However, industrial food waste often comes from the parts of plants that humans usually do not eat. Thus, food made from this waste is considered a novel food, which requires safety assessment for consumption in humans [23,24]. In this study, we reported a randomized placebo-controlled trial to assess the clinical safety and potential efficacy of continuously repeated intake of coffee cherry pulp juice concentrate, a product made from food waste from coffee manufacturing. The result demonstrates that repeated intake of coffee cherry pulp juice concentrate of 28 g/day over 12 weeks can be considered safe in healthy volunteers. Consuming coffee cherry pulp juice concentrate for 12 weeks resulted in no adverse symptoms and no differences in the mean changes in body weight, kidney function markers (BUN), liver function markers (ALT), hematology parameters, urinalysis parameters, and electrocardiogram characteristics from those of the placebo control group. Interestingly, continuous intake of coffee cherry pulp juice concentrate for at least 8 weeks significantly decreased cholesterol and LDL levels. The insight from this work can be used to design further clinical trials for the efficacy study of coffee cherry pulp products, in which a dose of coffee cherry pulp juice concentrate of up to 28 g per day can be used.

Previous studies have reported the safety of whole coffee fruit products [13,14]. However, this is the first study to report the long-term clinical safety of a coffee cherry pulp product. Among all parts of the coffee fruit, the coffee pulp contains an abundant level of phytochemicals with antioxidant properties [25]. Compared to the results of previous studies in whole fruit products, this study showed a higher safe dose level for coffee cherry pulp juice concentrate. A previous study reported that consuming 800 mg/day of whole coffee fruit powder for 28 days was safe for college athletes [14]. Generally Recognized as Safe (GRAS) Notice (GRN) No. 868 allowed the use of Coffeeberry^®^ Coffee Fruit Extract at levels of up to 300 mg/serving [13]. The reason why the safe dose of coffee cherry pulp juice concentrate in our study (28 g per day) is higher than those of the other whole coffee fruit products is likely due to the fact that the coffee pulp product excludes the coffee seed or bean and contains a very low amount of caffeine (only 0.38%). Furthermore, the coffee cherry pulp contains a number of phytochemicals with antioxidant properties [26]. The total antioxidant activity measured by ORAC assay is 9947.13–10,995.13 μmoles TE per 100 g of product, which may counteract some deleterious effects and help maintain blood parameter values in a normal range. Interestingly, a recently published work used a risk assessment to estimate the safe intake level of coffee cherry products based on the safe dose of caffeine, epigallocatechin gallate, and trigonelline [27]. The results showed that the amount of coffee cherry pulp and husk products that can be safely consumed at one time was much higher than the average amounts expected to be consumed by the population [27]; these results are consistent with our findings in this study.

During the study, participants who received the coffee cherry pulp juice concentrate maintained fasting plasma glucose levels significantly better than the placebo group. The finding suggests that coffee cherry pulp juice concentrate may offer some benefits in controlling fasting plasma glucose levels. In addition, a significant decrease in total cholesterol and LDL was observed after consuming coffee cherry pulp juice concentrate for at least 8 weeks, with a significant difference from the placebo group. Since some participants in this study had total cholesterol and LDL higher than normal levels (but still had a normal cholesterol:HDL ratio), the findings from this study indicate some beneficial effects of the product on blood lipid control. This result is consistent with our previous short-term study. We found that consuming 20–28 g/day of coffee cherry pulp juice concentrate for 2 weeks significantly reduced total cholesterol and LDL cholesterol [18]. An increase in LDL cholesterol and total cholesterol are biomarkers of metabolic-related dyslipidemia, which can increase the risks of obesity and hypertension, diabetes mellitus, cardiovascular disease (CVD), and hypothyroidism [28,29]. Thus, the efficacy of coffee cherry pulp juice concentrate in blood lipid control could have potential application in preventing dyslipidemia and other related diseases. Interestingly, some animal studies using coffee pulp aqueous extract (CPE) also showed benefits in modulating glucose intolerance and hyperlipidemia [8,30,31,32]. The mechanisms of controlling blood sugar and lipids are likely through the inhibition of intestinal cholesterol absorption [8] and antioxidative effects from polyphenols such as chlorogenic acid (CGA), epicatechin (EC), and catechin [30,31]. Interestingly, a previous in vitro simulated gastrointestinal digestion study reported that the phenolic acids and caffeine in coffee pulp extract were more bioaccessible than flavonoids [33]. Thus, phenolic acids are likely to contribute to the antioxidant activities and the blood-lipid-lowering effect of coffee cherry pulp juice concentrate. Future clinical trials to study the efficacy of coffee cherry pulp juice that concentrate on blood sugar and lipid in people with prediabetes and dyslipidemia are worthwhile.

Compared to the placebo group, a significant increase in the average levels of HDL and triglycerides (TG) was observed at week 14, which was 2 weeks after stopping the interventions. The slight increase in HDL and TG in the study group was not significant when compared to the baseline values (Figure 6A,C). However, the placebo group did show a slight decrease in HDL and TG levels (Figure 6B,D). Therefore, the significance between groups is more likely to be caused by the slight increase in HDL in the study group and the slight decrease in HDL in the control group. The decrease in HDL in the placebo group after stopping the intervention may be due to other factors, such as reduced physical activity [34]. Although we asked all participants to maintain their routine exercises throughout the study, there could be some changes in their physical movement from their jobs or housework. The main function of HDL is to export cholesterol and other lipids from peripheral tissues (such as the cardiovascular system) to the liver for disposal, to steroidogenic tissues for hormone production, and to exchange lipids with apoB-containing particles [35]. Though epidemiological studies found an inverse association between HDL levels and cardiovascular risk [36], a recent meta-analysis showed no improvement in cardiovascular outcomes after elevating HDL-C levels [37]. Recent evidence suggests that the quality of HDL is more important than quantity [34,35]. For triglycerides, it is worth noting that the increased average level of TG observed in the study group on week 14 was still within the normal range (lower than 150 mg/dL). Over the course of the intervention, there were no changes in triglycerides. Therefore, the increased TG is likely not related to the coffee cherry pulp juice concentrate but rather to other causes. The most common contributor to the increase in triglyceride is the intake of high-calorie food [38]. Although we asked all participants to maintain their eating behavior and record their diet three days per week throughout the study, there could be some changes in diet intake on the days that were not on the records.

Compared to the placebo group, a significant decrease in the average levels of creatinine and AST was observed at week 14 (two weeks after stopping the interventions). The slight decrease in creatinine and AST in the study group was not significant when compared to the baseline values (Figure 7C and Figure 8C). However, the placebo group did show a slight increase in creatinine and AST levels (Figure 7D and Figure 8D). Therefore, the significance between groups is more likely due to both the slight increase in HDL in the study group and the slight decrease in HDL in the control group. Increased creatinine and AST have been used in the diagnosis of kidney and liver injuries, respectively [39,40]. Therefore, the observed decrease in creatinine and AST is considered a favorable outcome. Nevertheless, from this study in healthy volunteers, we can only conclude that coffee cherry pulp juice concentrate at a dose of 28 g/day is not toxic to the kidneys or liver. Further studies in people with abnormal levels of kidney and liver biomarkers are warranted to investigate the efficacy of coffee pulp products in improving kidney and liver function.

The first strength of this work is its randomized blinded placebo control trial design, which has low bias. Second, the dose of 28 g/day was designed based on our previous dose-escalation study of two weeks, which helped lower the risk to participants. Third, all participants were followed up periodically at 4, 8, and 12 weeks of intervention, and 2 weeks after stopping the intervention. The multiple follow-ups allow for careful monitoring of safety and help maintain good compliance. The limitations of this study include the difficulty of avoiding menstrual periods on the follow-up dates. Thus, several participants had slightly cloudy urine, RBC, WBC, and squamous epithelial cells on the appointment date. Second, our inclusion criteria only required normal blood pressure and heart rate but did not require normal EKG. Thus, in this study, there were also some participants with abnormal EKG in both groups at baseline. For this study, consuming coffee cherry pulp juice concentrate had no adverse effects on EKG. However, future clinical safety studies should set normal EKG as an inclusion criteria.

## 5. Conclusions

The findings of this study suggest that a continuous intake of coffee cherry pulp juice concentrate of 28 g per day over a period of 12 weeks is considered safe for healthy volunteers. Moreover, it may provide beneficial effects in maintaining fasting plasma glucose and reducing total cholesterol and LDL. Future clinical trials for the efficacy study of the coffee cherry pulp products can use a dose of up to 28 g per day.

## Figures and Tables

**Figure 1 nutrients-15-01602-f001:**
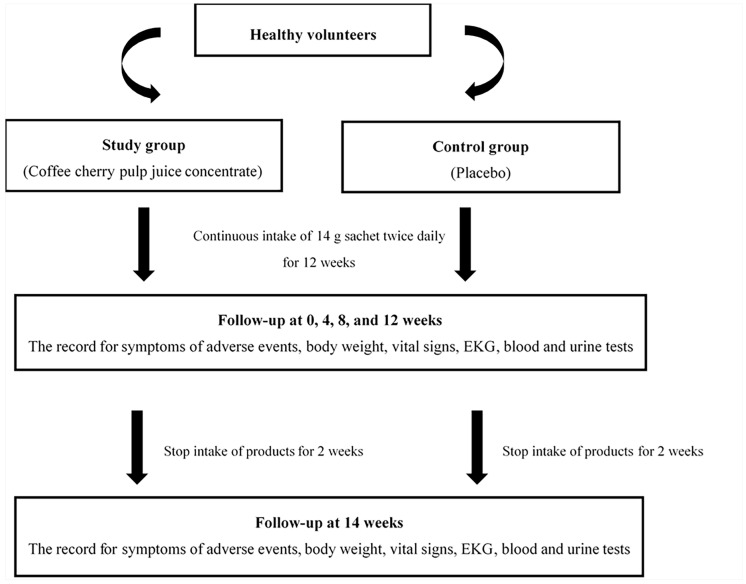
Conceptual framework.

**Figure 2 nutrients-15-01602-f002:**
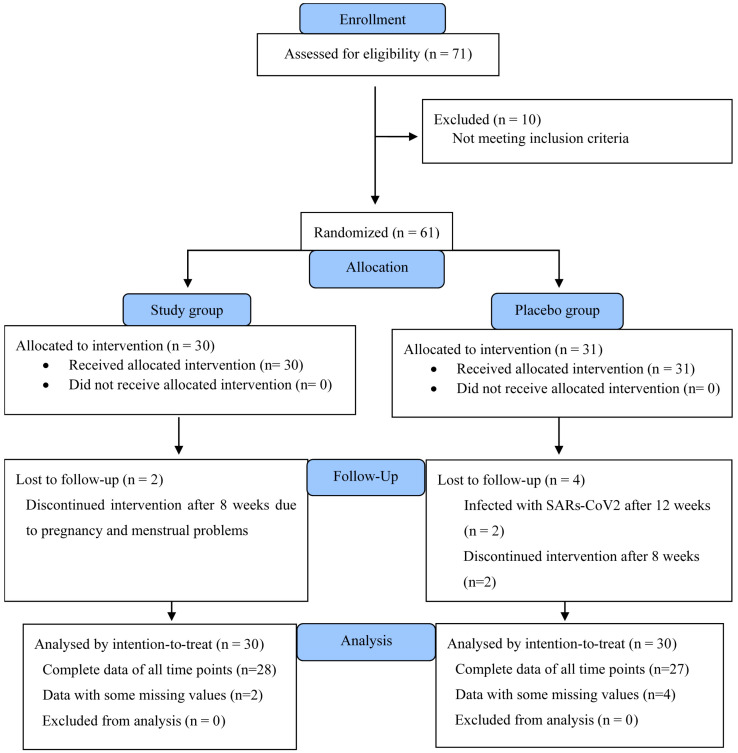
CONSORT participant flowchart.

**Figure 3 nutrients-15-01602-f003:**
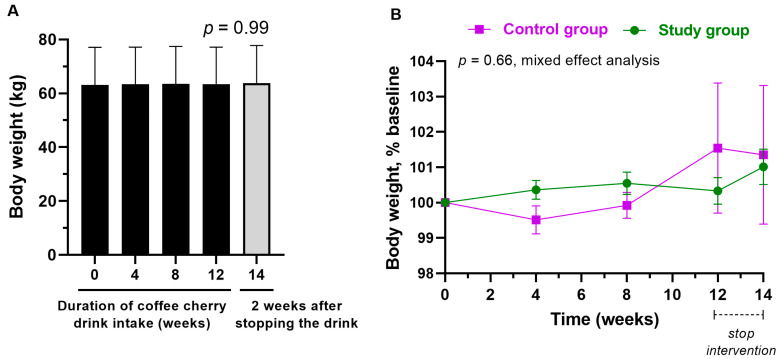
Changes in body weight of the volunteers. (**A**) The bar graph shows the mean and standard deviation of body weight in the study group at 0, 4, 8, 12, and 14 weeks, with *p*-value obtained from the Kruskal–Wallis test; (**B**) the line graph shows the mean and standard deviation of body weight (% baseline) in the study (green) and control (purple) groups, with *p*-value obtained from mixed effect analysis.

**Figure 4 nutrients-15-01602-f004:**
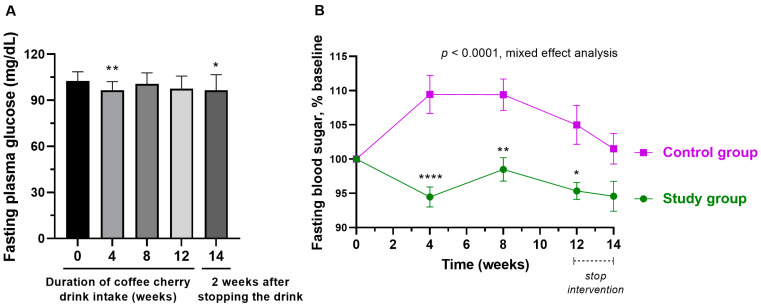
Changes in fasting plasma glucose of the volunteers. (**A**) The bar graph shows the mean and standard deviation of the volunteer’s fasting plasma glucose after the cherry coffee concentrate intake for 4, 8, and 12 weeks and 2 weeks after stopping intake (14 weeks), *p*-value analyzed by the Kruskal–Wallis test; (**B**) the line graph shows the mean and standard deviation of the volunteer’s fasting plasma glucose after the coffee cherry pulp juice concentrate and placebo intake compared to % baseline, *p*-value analyzed by mixed-effect analysis. *, **, and **** mean *p* < 0.05. 0.01, and 0.0001, respectively.

**Figure 5 nutrients-15-01602-f005:**
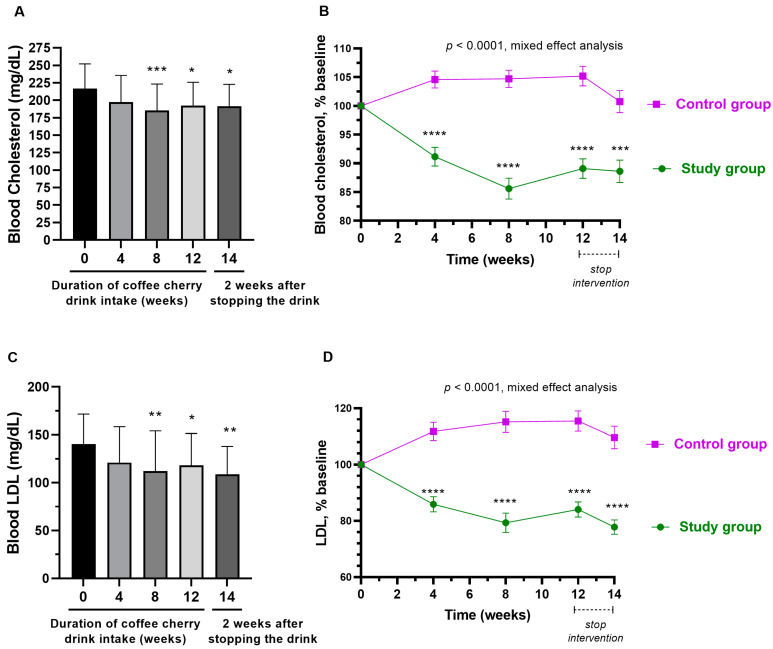
Changes in total cholesterol and LDL. The bar graphs show the mean and standard deviation of total cholesterol (**A**) and LDL (**C**) after consumption of cherry coffee pulp juice concentrate for 4, 8, and 12 weeks, and 2 weeks after stopping the intervention (14 weeks), with *p*-value obtained from the Kruskal–Wallis test; the line graphs show a comparison of the mean and standard deviation of the total cholesterol (**B**) and LDL (**D**) in the percentage of the respective baseline values (% baseline) at each time point between study (green) and control (purple) groups, with *p*-value obtained from mixed-effect analysis. *, **, ***, and **** mean *p* < 0.05. 0.01, 0.001, and 0.0001, respectively.

**Figure 6 nutrients-15-01602-f006:**
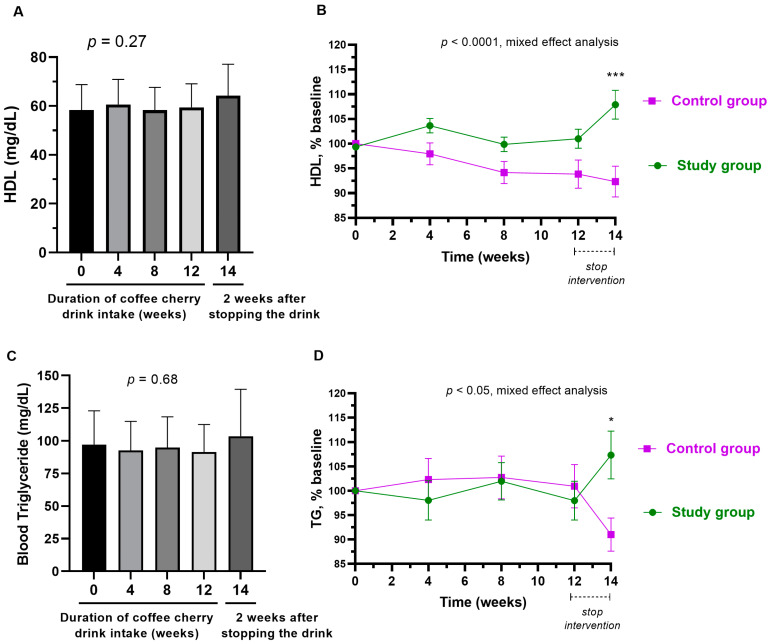
Changes in HDL and triglyceride. The bar graphs show the mean and standard deviation of HDL (**A**) and triglyceride (**C**) after consumption of cherry coffee pulp juice concentrate for 4, 8, and 12 weeks, and 2 weeks after stopping the intervention (14 weeks), with *p*-value obtained from the Kruskal–Wallis test; the line graphs show a comparison of the mean and standard deviation of the HDL (**B**) and triglyceride (**D**) in the percentage of the respective baseline values (% baseline) at each time point between study (green) and control (purple) groups, with *p*-value obtained from mixed-effect analysis. *, and *** mean *p* < 0.05, and 0.001, respectively.

**Figure 7 nutrients-15-01602-f007:**
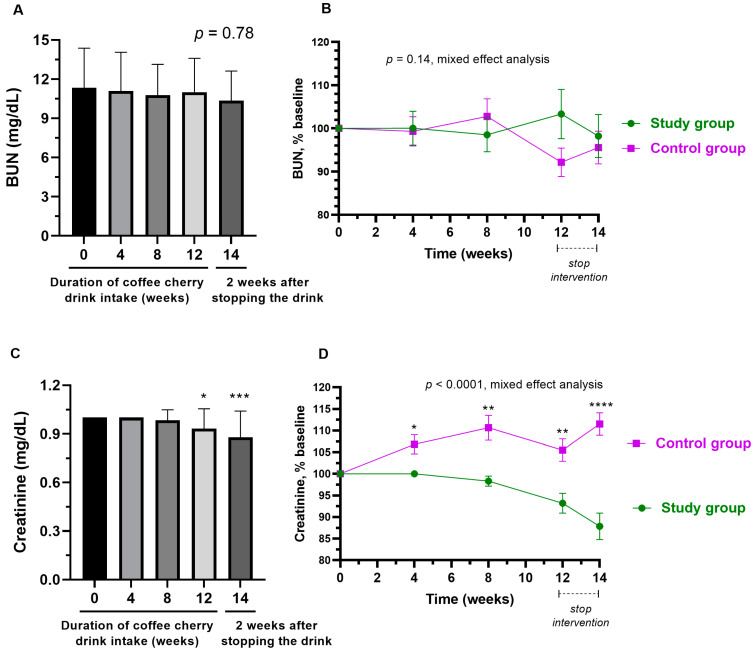
Changes in BUN and creatinine. The bar graphs show the mean and standard deviation of BUN (**A**) and creatinine (**C**) after consumption of cherry coffee pulp juice concentrate for 4, 8, and 12 weeks, and 2 weeks after stopping the intervention (14 weeks), with *p*-value obtained from the Kruskal–Wallis test; the line graphs show a comparison of the mean and standard deviation of the BUN (**B**) and creatinine (**D**) in the percentage of the respective baseline values (% baseline) at each time point between study (green) and control (purple) groups, with *p*-value obtained from mixed-effect analysis. *, **, ***, and **** mean *p* < 0.05. 0.01, 0.001 and 0.0001, respectively.

**Figure 8 nutrients-15-01602-f008:**
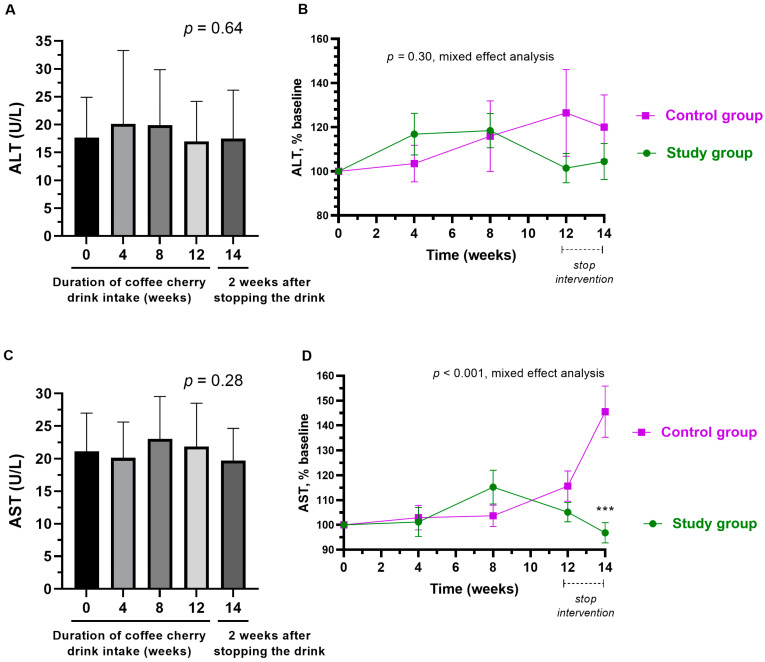
Changes in ALT and AST. The bar graphs show the mean and standard deviation of ALT(**A**) and AST (**C**) after consumption of cherry coffee pulp juice concentrate for 4, 8, and 12 weeks, and 2 weeks after stopping the intervention (14 weeks), with *p*-value obtained from the Kruskal–Wallis test; the line graphs show a comparison of the mean and standard deviation of the ALT (**B**) and AST (**D**) in the percentage of the respective baseline values (% baseline) at each time point between study (green) and control (purple) groups, with *p*-value obtained from mixed-effect analysis. *** means *p* < 0.001.

**Figure 9 nutrients-15-01602-f009:**
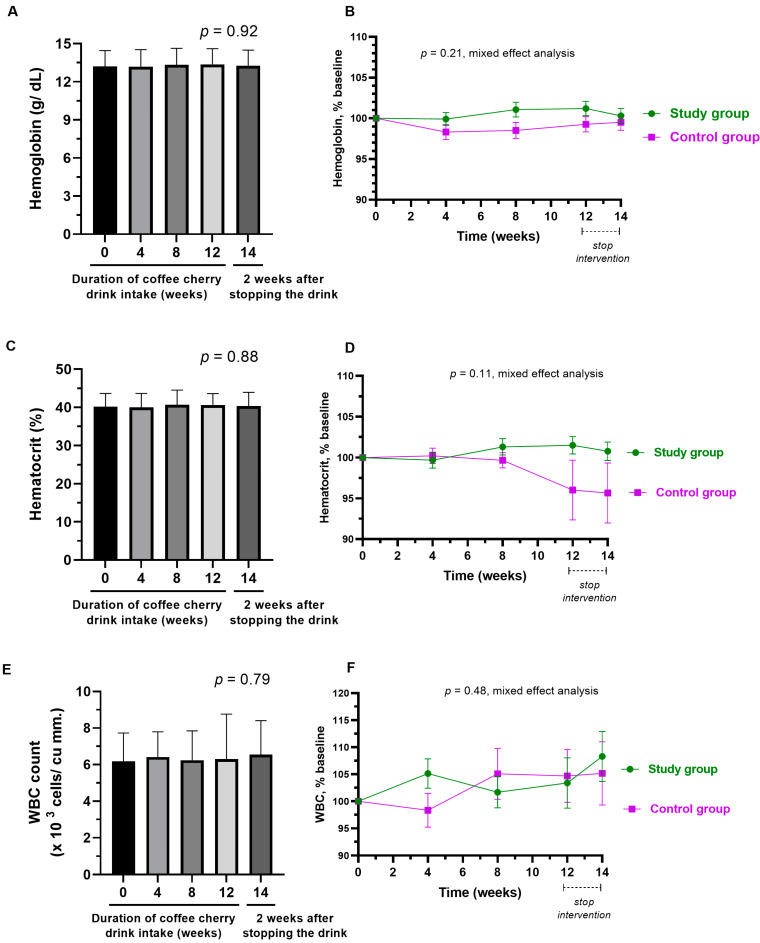
Changes in hemoglobin, hematocrit, and WBC. The bar graphs show the mean and standard deviation of hemoglobin (**A**), hematocrit (**C**), and WBC (**E**) after consumption of cherry coffee pulp juice concentrate for 4, 8, and 12 weeks, and 2 weeks after stopping the intervention (14 weeks), with *p*-value obtained from the Kruskal–Wallis test; the line graphs show a comparison of the mean and standard deviation of the hemoglobin (**B**), hematocrit (**D**), and WBC (**F**) in the percentage of the respective baseline values (% baseline) at each time point between study (green) and control (purple) groups, with *p*-value obtained from mixed-effect analysis.

**Figure 10 nutrients-15-01602-f010:**
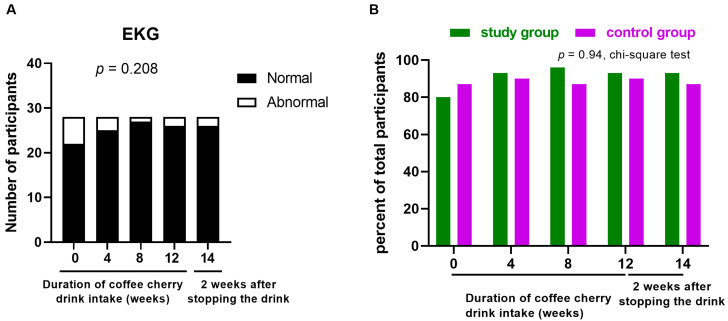
Changes in EKG. The stacked bar graph shows the number of participants with normal (black bars) and abnormal (white bars) EKG characteristics after the cherry coffee concentrate intake for the specified period and 2 weeks after stopping intake (14 weeks) (**A**), with *p*-value obtained from chi-squared test; the bar graph shows the percentage of participants with normal EKG characteristics in the study (green) and control (purple) groups (**B**).

**Table 1 nutrients-15-01602-t001:** Demographic characteristics of the participants at baseline.

	**Study Group (*n* = 30)**	**Control Group** **(*n* = 31)**	***p*-Value ^(1)^**
Parameter	N (%)	N (%)	
**Gender**			
Female	22 (73.33)	21 (67.74)	0.78 ^(1)^
Male	8 (26.67)	10 (32.26)	
**Systemic diseases**			
None Allergic rhinitis	27 3	29 2	0.67
**Smoking**	0	0	>0.99
**Alcohol drinking**	1 (3.33)	2 (6.45)	>0.99
	**Study Group** **(*n* = 30)**	**Control Group (*n* = 31)**	***p*-Value ^(2)^**
**Parameter**	Mean ± SD	Mean ± SD	
Age (years)	32.43 ± 6.82	30.87 ± 7.39	0.42
BMI (kg/m^2^)	23.7 ± 4.48	23.09 ± 4.13	0.49
Systolic blood pressure (mmHg)	116.93 ± 11.33	118.2 ± 14.1	0.72
Diastolic blood pressure (mmHg)	78.57 ± 10.98	74.9 ± 9.8	0.17
Pulse rate (beats/min)	77.57 ± 8.48	74.43 ± 12.98	0.28

Abbreviations: BMI, body mass index; *n* = the number of participants. *p*-values were obtained from ^(1)^ Fisher’s exact test, or ^(2)^ unpaired *t*-tests.

**Table 2 nutrients-15-01602-t002:** Clinical hematological and blood chemistry values of the participants at baseline.

Parameter	Study Group (*n* = 30)	Control Group (*n* = 31)	*p*-Value
WBC (cells/cu.mm)	6.37 ± 1.45 × 10^3^	6.34 ± 1.18 × 10^3^	0.55
RBC (million cells/cu.mm)	4.83 ± 0.43 × 10^6^	4.56 ± 0.35 × 10^6^	0.42
Hemoglobin (g/dL)	13.46 ± 0.95	13.31 ± 1.19	0.89
Hematocrit value (%)	40.82 ± 3.2	40.69 ± 3.32	0.96
MCV (fL)	86.07 ± 3.83	84.6 ± 6.94	0.27
MCH (pg)	29.01 ± 1.46	27.9 ± 2.72	0.30
MCHC (g/dL)	33.51 ± 0.42	33 ± 0.82	0.69
RBDW (%)	13.15 ± 0.84	12.89 ± 0.96	0.06
Platelet count (cells/cu.mm)	290,900 ± 74,118.89	279,709.68 ± 63,507.59	0.46
Neutrophils (%)	55.37 ± 8.45	44.94 ± 12.17	0.43
Lymphocytes (%)	32.54 ± 6.22	32.18 ± 20.44	0.28
Monocytes (%)	6.13 ± 1.23	15.23 ± 15.49	0.53
Eosinophil (%)	2.64 ± 1.77	3.54 ± 3.41	0.24
Basophil (%)	0.77 ± 0.43	1.56 ± 1.66	0.59
Fasting plasma glucose (mg/dL)	104.33 ± 6.91	90.68 ± 10.14	***
HbA1C (%)	5.27 ± 0.31	5.10 ± 0.41	0.08
Total cholesterol (mg/dL)	218 ± 35.00	194.30 ± 28.01	**
Triglyceride (mg/dL)	100.3 ± 32.49	97.42 ± 26.07	0.70
HDL cholesterol (mg/dL)	58.07 ± 11.23	68.00 ± 13.00	**
LDL cholesterol (mg/dL)	141.03 ± 30.00	105.89 ± 24.04	****
BUN (mg/dL)	11.37 ± 2.92	11.21 ± 2.98	0.73
Creatinine (mg/dL)	0.64 ± 0.11	0.71 ± 0.09	***
eGFR (mL/min/1.73 m^2^)	121.63 ± 8.85	110.75 ± 10.62	***
AST (U/L)	21.20 ± 5.87	18.33 ± 4.79	*
ALT (U/L)	18.87 ± 8.85	15.57 ± 8.76	*
Total bilirubin (mg/dL)	0.52 ± 0.12	0.44 ± 0.23	0.09
Sodium level (mg/dL)	136.6 ± 2.49	134.33 ± 6.66	0.08
Potassium level (mg/dL)	4.14 ± 0.28	4.16 ± 0.35	0.61
Chloride level (mg/dL)	100.27 ± 1.66	100.41 ± 1.96	0.93
Carbon dioxide level (mg/dL)	19.53 ± 2.10	20.27 ± 1.39	0.13

Data are shown as mean ± SD. Abbreviations: WBC, white blood cell count; RBC, red blood cell count; MCV, mean cell volume; MCH, mean cell hemoglobin; MCHC, mean cell hemoglobin concentration; RDW, red blood cell distribution width; HBA1C, glycosylated hemoglobin; HDL, high-density lipoprotein; LDL, low-density lipoprotein; BUN, blood urea nitrogen; eGFR, estimated glomerular filtration rate; AST, aspartate transaminase; ALT, alanine aminotransferase; *p*-values were obtained from unpaired *t*-test. *, **, ***, and **** represent *p* < 0.05, 0.01, 0.001, and 0.0001, respectively.

**Table 3 nutrients-15-01602-t003:** Summary of adverse symptoms.

Variables	Study Group (*n* = 30)	Control Group (*n* = 31)
Participants with any AEs	0	0
Specific AEs	0	0
Burning mouth	0	0
Nausea	0	0
Vomiting	0	0
Abdominal pain	0	0
Headache	0	0

Data are shown as the number of participants with adverse symptoms.

**Table 4 nutrients-15-01602-t004:** Number of participants with abnormal urinalysis parameters.

Parameter	Normal Range	Study Group (*n* = 30)					Placebo Group (*n* = 31)				
		Baseline	Week 4	Week 8	Week 12	Week 14	Baseline	Week 4	Week 8	Week 12	Week 14
Color	Colorless	0	0	0	0	0	0	0	0	0	0
Appearance	Clear	0	0	0	0	0	0	0	0	0	0
Specific gravity	1.003–1.030	0	0	0	0	0	0	0	0	0	0
pH	5.0–8.0	0	0	0	0	0	0	0	0	0	0
Leukocyte	Negative	0	0	0	0	0	0	0	0	0	0
Nitrite	Negative	0	0	0	0	0	0	0	0	0	0
Glucose	Negative	0	0	0	0	0	0	0	0	0	0
Protein	Negative	1 (1+)	0	1 (2+)	0	1	1 (2+)	0	0	0	0
Bilirubin	Negative	0	0	0	0	0	0	0	0	0	0
Ketone	Negative	0	0	0	0	0	0	0	0	0	0
Urobilinogen	Normal	0	0	0	0	0	0	0	0	0	0
Blood	Negative	0	1 (4+)	0	0	0	1 (4+)	2 (4+, 4+)	0	0	0
WBC	0–1 cell/HP	0	1 (2–3)	0	0	0	1 (2–3)	2 (2–3, 2–3)	0	0	0
RBC	0–1 cell/HP	0	2 (3–5, 3–5)	0	0	1 (3–5)	1 (3–5)	2 (3–5, 3–5)	0	0	0
Squamous epithelial cell	0–1 cell/HP	1 (3–5)	2 (3–5, 3–5)	0	0	0	0	2 (3–5, 3–5)	0	0	0
Bacteria	Few	0	0	0	0	0	0	0	0	0	0

Data are expressed as the number of participants with abnormal values, *n*, (values) Abbreviations: WBC, white blood cell; RBC, red blood cell, HP, high power field.

## Data Availability

All data collected in this project have been described in this work. Since the participants only gave consent to report the summary of data, no individual data can be shared publicly. However, if any readers would like to view anonymized individual data, please send an e-mail to the corresponding author.

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
