# Peer review of "Twelve-Week Safety and Potential Lipid Control Efficacy of Coffee Cherry Pulp Juice Concentrate in Healthy Volunteers"

_nutrients, 2023, doi:10.3390/nu15071602_

Round 1

Reviewer 1 Report

The authors described an interesting and timely topic: the use of agricultural by-products to increase sustainability.

The following revisions are necessary:

Throughout: a major revision of English language and grammar is necessary, as well as major copy-editing for style and clarity. The text is difficult to read and contains many mistakes.

Throughout: the text inside the figures should also be language edited (“compare the …” is strange). All figures should be aligned for layout (bar sizes etc). Figure 10b misses axes.

Introduction:

Line 36: what FDA? Is this the US FDA? How is this relevant for novel food approval in the EU and/or Thailand?

Line 78: was the taste of the products the same? How did you reconstruct the coffee cherry taste for the placebo. Information about intake of the products is also missing. Were they reconstituted in water before use? Or was the product ingested within the sachet?

Section 2.4. Please provide manufacturer and citation for G-power software. Please provide exact details and results as supplementary information. Do you mean effect size index value?

Section 2.5. Please state species and variety of Coffee type used.

Line 440: it is probably untrue that all data are described in the work. The figures only show the percentage data. Please provide tables of all numerical data as data supplement.

Discussion: an important reference was overlooked: Eckhardt et al. Risk Assessment of Coffee Cherry (Cascara) Fruit Products for Flour Replacement and Other Alternative Food Uses. Molecules. 2022; 27(23):8435. https://doi.org/10.3390/molecules27238435. Basically this references independently validates the conclusions of  the study using another methodology.

Author Response

The authors described an interesting and timely topic: the use of agricultural by-products to increase sustainability. The following revisions are necessary:

  1. Throughout a major revision of English language and grammar is necessary, as well as major copy-editing for style and clarity. The text is difficult to read and contains many mistakes.

Response: Thank you for your suggestion. We checked and edited the grammatical use of the manuscript as suggested. Also, we have the language professionally checked and edited by the MDPI service. The certificate of language check is attached with the revised submission.   

  1. Throughout the text inside the figures should also be language edited (“compare the …” is strange). All figures should be aligned for layout (bar sizes etc). Figure 10b misses axes.

Response: Thank you for your advice. We corrected the figures as suggested.

Introduction:

  1. Line 36: what FDA? Is this the US FDA? How is this relevant for novel food approval in the EU and/or Thailand?

Response: Thank you for your comment. We revised the text to “the regulatory organizations such as The European Food Safety Authority (EFSA) or Thai Food and Drug Administration (Thai FDA)” as shown on page 1 lines 35-37.

  1. Line 78: was the taste of the products the same? How did you reconstruct the coffee cherry taste for the placebo. Information about intake of the products is also missing. Were they reconstituted in water before use? Or was the product ingested within the sachet?

Response: Thank you for your question. As additionally described in Method on pages 3-4

Lines 136-139: The placebo was made using water, fructose syrup, sodium citrate, malic acid, brown col-or powder, and carboxymethylcellulose to match the coffee cherry pulp juice concentrate for weight, taste, color, flavor, and acidity (equal pH of 4.5).

Lines 142-146: To ensure the similarity between the placebo and the juice, a group of trained panelists tasted both products and confirmed that the taste of a placebo product was similar to that of the juice concentrate, which is sweet and sour (which resembles prune juice). Also, the brix and pH were measured to ensure the similar property of both products.

Lines 151-155: The sachets of both coffee cherry pulp juice concentrate and placebo are ready-to-eat. All participants were asked to consume the products directly from the sachets by using the given disposable straws. They were instructed to record their daily consumption of the assigned products in the subject diaries.

  1. Section 2.4. Please provide manufacturer and citation for G-power software. Please provide exact details and results as supplementary information. Do you mean the effect size index value?

Response: On page 3 lines 102-106, we added statements to describe the sample size calculation, add a reference, and provide the captured picture of the calculation in the supplementary information as suggested.

The sample size of the present study was calculated by using G-power V.3.1.9.4 (Heinrich-Heine-Universität Düsseldorf, Düsseldorf, Germany) [21]. The theoretical large effect size value of 0.8 for the comparison of two independent means (the unpaired t-test) was used. With a power of 0.8 and a significance level of 0.05, the calculation of sample size yielded the result of at least 26 people per group (Supplementary material). Assuming a dropout rate of 15%, the sample size was set at thirty people (n = 30) in each group, and the total sample size was 60.

  1. Section 2.5. Please state species and variety of Coffee type used.

Response: Coffee Arabica L variety Catimor was added on page 3 line 121.

  1. Line 440: it is probably untrue that all data are described in the work. The figures only show the percentage data. Please provide tables of all numerical data as a data supplement.

Response: Thank you for your comment. We only obtained written informed consent for publishing a summary of the data not the individual data of each subject. To address your concern, as shown on page 20 lines 508-511 we revised the statement to

All data collected in this project have been described in this work. Since the participants only gave consent to report the summary of data, no individual data can be shared in public. However, if any readers want to see de-identifiable individual data, please send an e-mail to the corresponding author.

  1. Discussion: an important reference was overlooked: Eckhardt et al. Risk Assessment of Coffee Cherry (Cascara) Fruit Products for Flour Replacement and Other Alternative Food Uses. Molecules. 2022; 27(23):8435. https://doi.org/10.3390/molecules27238435. Basically this references independently validates the conclusions of the study using another methodology.

Response: Thank you for the useful suggestion. We included the reference and statements to describe this work in the discussion on page 19 lines 407-412 as follows.

Interestingly, a recently published work used a risk assessment to estimate the safe intake level of coffee cherry products based on the safe dose of caffeine, epigallocatechin gallate, and trigonelline [27]. The result showed that the amount of coffee cherry pulp and husk products that can be safely consumed at one time was much higher than the average amounts expected to be consumed by the population [27]. Thus, the result of the previous study is consistent with our findings in this study.

Reviewer 2 Report

In the manuscript ‘Repeated dose studies for Clinical Safety and Potential Efficacy of Coffee Cherry Pulp Juice Concentrate in Healthy Volunteers’ authors evaluated the safety and potential efficacy of coffee cherry pulp juice concentrate.  Although the manuscript is well written, neat, and the material and methods well explained, in the reviewer's opinion, the manuscript needs minor revisions.

-          The title does not detail the conclusions of the article and should be modified.

-          Line 36: FDA must be detailed.

-          Line 83: the inclusion criteria of total cholesterol is missing.

-          Lines 97-98: The sentence must be checked

-          Line 109: 126 – 139 mg eq GA : can confuse Gallic acid? or chlorogenic acid?

-          Line 123: Per 100 g of the product, there is 0.29 - 0.33% caffeine, 2.7- 3.00 g dietary fiber, 3.3 - 3.7 g protein, 0.18 - 0.20 mg  vitamin E, 34.2 – 37.8 g sugar, 0.17 – 0.19 mg zinc, 0.716 - 0.796 mg copper. There is still no information about the 100 g of product.

-          Line 182: There were 71 subjects initially screened for eligibility and thirty-one of them passed the screening. 31 or 61?

-          Tables: I advise authors to change the structure of the table: First I would put the control column.

-          Lines 259-260: The sentence must be checked.

-          The increase in HDL and triglycerides after 14 weeks should be discussed.

-          The increase in creatinine and AST after 14 weeks should be discussed.

-          In my opinion, the discussion should be widely improved.

Author Response

In the manuscript ‘Repeated dose studies for Clinical Safety and Potential Efficacy of Coffee Cherry Pulp Juice Concentrate in Healthy Volunteers’ authors evaluated the safety and potential efficacy of coffee cherry pulp juice concentrate.  Although the manuscript is well written, neat, and the material and methods well explained, in the reviewer's opinion, the manuscript needs minor revisions.

  1. The title does not detail the conclusions of the article and should be modified.

Response: Thank you for your suggestion. We revise the title to “12-weeks Safety and Potential lipid control Efficacy of Coffee Cherry Pulp Juice Concentrate in Healthy Volunteers”, as shown on page 1 lines 2-3.

  1. Line 36: FDA must be detailed.

Response: Thank you for your comment. We revised the text to “the regulatory organizations such as The European Food Safety Authority (EFSA) or Thai Food and Drug Administration (Thai FDA)”, as shown on page 1 lines 35-37

  1. Line 83: the inclusion criteria of total cholesterol is missing.

Response: Thank you for your suggestion. To address the issue, we added the following statements on pages 2-3 lines 86-90 as follows.

Blood cholesterol > 200 mg/dL is common even in healthy populations. To reflect the status of the general population and ensure the feasibility of this study, we included participants with total cholesterol ≤ 280 mg/dL and with total cholesterol: HDL ratio ≤ 5:1. The cut-off values were set based on previous studies for significantly lower risk of mortality in the general population [19-20].

  1. Lines 97-98: The sentence must be checked

Response: Thank you for your comment. As suggested by two reviewers, we checked and revised the text on page 3 lines 102-106 as follows.

The sample size of the present study was calculated by using G-power V.3.1.9.4 (Heinrich-Heine-Universität Düsseldorf, Düsseldorf, Germany) [21]. The theoretical large effect size value of 0.8 for the comparison of two independent means (the unpaired t-test) was used. With a power of 0.8 and a significance level of 0.05, the calculation of sample size yielded the result of at least 26 people per group (Supplementary material). Assuming a dropout rate of 15%, the sample size was set at thirty people (n = 30) in each group, and the total sample size was 60.

  1. Line 109: 126 – 139 mg eq GA: can confuse Gallic acid? or chlorogenic acid?

Response: As shown on page 3 line 114, we provided a full description of mg eq GA as milligram gallic acid equivalents.

  1. Line 123: Per 100 g of the product, there is 0.29 - 0.33% caffeine, 2.7- 3.00 g dietary fiber, 3.3 - 3.7 g protein, 0.18 - 0.20 mg vitamin E, 34.2 – 37.8 g sugar, 0.17 – 0.19 mg zinc, 0.716 - 0.796 mg copper. There is still no information about the 100 g of product.

Response: We described the composition of nutrients per 100 g product based on Thai law for the nutrient claims that require the nutrient values based on 100 g product for a new type of food. The amount of bioactive compound in a 14 g sachet has been described on page 3 lines 113-117.

  1. Line 182: There were 71 subjects initially screened for eligibility and thirty-one of them passed the screening. 31 or 61?

Response: Sorry for the mistakes. Yes. It should be sixty-one of them passed the screening. We revised it as appeared on page 5 line 202.

  1. Tables: I advise authors to change the structure of the table: First I would put the control column.

Response: Thank you for your advice. In this study, we focus on the safety and efficacy of the coffee cherry pulp juice concentrate product. Therefore, we prefer to show the data of the study group first and then compared it with the control group to ensure that the effect was not a placebo effect.

  1. Lines 259-260: The sentence must be checked.

Response: Thank you for your suggestion. We checked and revise the text on page 10 lines 277 and 279 as follows.

Similarly, the average level of LDL was significantly decreased after 8- and 12-week consumption of coffee cherry pulp juice concentrate (p < 0.01 and p < 0.05, respectively). Also, the average level of LDL was still significantly lower than the baseline even after stopping the intervention for 2 weeks (p < 0.01) (Figure 259 5C). No significant change was observed in the placebo group. Compared to the placebo group, the study group had significantly reduced levels of LDL starting from week 4 (p<0.0001) (Figure 5D).

  1. The increase in HDL and triglycerides after 14 weeks should be discussed.

Response: Thank you for the advice. We add statements in the discussion on pages 19-20 lines 439-463.

  1. The increase in creatinine and AST after 14 weeks should be discussed.

Response: We add statements in the discussion on page 20 lines 464-476.  

  1. In my opinion, the discussion should be widely improved.

Response: Thank you for your suggestion. We revised the discussion part as recommended. More statements were added on pages 18-19 lines 407-412, 423-428, 439-476.